# Somatotype Components as Useful Predictors of Disordered Eating Attitudes in Young Female Ballet Dance Students

**DOI:** 10.3390/jcm9072024

**Published:** 2020-06-27

**Authors:** José Ramón Alvero-Cruz, Verónica Parent Mathias, Jerónimo C. García-Romero

**Affiliations:** 1Department of Human Physiology, Histology, Pathological Anatomy and Physical Education and Sport, University of Málaga-Andalucía Tech, 29071 Málaga, Spain; veronicaparent@hotmail.com (V.P.M.); jeronimo@uma.es (J.C.G.-R.); 2The Biomedical Research Institute of Málaga (IBIMA), 29010 Málaga, Spain; 3Edificio López de Peñalver, Campus de Teatinos, Universidad de Málaga, 29071 Málaga, Spain

**Keywords:** dance students, disordered eating attitudes, Eating Attitudes Test-26 (EAT-26), mesomorphy, ectomorphy, Receiver Operating Characteristics (ROC) curve analysis

## Abstract

The current study used receiver operating characteristic (ROC) curve analysis to examine the accuracy of somatotype components in correctly classifying disordered eating attitudes (DEA) in female dance students. Participants were a sample of 81 female dancers distributed in two groups: beginner training (BT; age (mean ± SD) = 10.09 ± 1.2 years, *n* = 32) and advanced training (AT; age = 15.37 ± 2.1 years, *n* = 49). For evaluation of DEA, the Eating Attitudes Test- 26 (EAT-26) questionnaire was used. We defined an EAT-26 score ≥20 as positive for DEA. Somatotype components were calculated using the Heath-Carter anthropometric method. The risk of presenting DEA was 28.1% (*n* = 9) in the BT group and 6.1% (*n* = 3) in the AT group. In the BT group, mesomorphy demonstrated moderate–high accuracy in predicting DEA (area under the curve (AUC) = 0.82, 95% confidence interval (CI): 0.64–0.93). The optimal cut-off of 6.34 yielded a sensitivity of 0.77 and a specificity of 0.95. Ectomorphy showed moderate accuracy in predicting DEA (AUC = 0.768, 95% CI: 0.58–0.89). The optimal cut-off of 2.41 yielded a sensitivity of 0.78 and a specificity of 0.78. In the AT group, none of the components demonstrated accuracy in predicting DEA. Somatotype components were good predictors of disordered eating attitudes in the younger dance student group (beginner training). Further research is needed to identify the determinants of these differences between the two groups.

## 1. Introduction

Eating disorders (EDs) are mental disorders defined by abnormal eating habits that negatively affect a person’s physical or mental health. Anorexia nervosa, one of the main EDs, has two distinct symptoms: low body weight (body mass index (BMI) less than 17.5 kg/m^2^ or less than 85% of the expected weight for height, age and sex) and body image disturbance. Bulimia nervosa is defined by three criteria: recurrent binge eating, recurrent compensatory behavior, and preoccupation with one’s body weight or shape. Others EDs include avoidant/restrictive food intake disorder, pica, regurgitation disorders, and other specified feeding and eating disorder [1]. In most EDs, the association with depression and anxiety states and substance abuse is common [1,2].

The influence of cultural and social factors on the development of EDs and their manifestations have been investigated from multiple perspectives [3]. The reasons for these rates of EDs have been centered on elements including personality factors and traits such as perfectionism and low self-esteem. Epigenetic reasons are currently found in the etiology of EDs, with evidence suggesting that epigenetic processes link malnutrition and life stresses (gestational, perinatal, childhood, and adult) to the risk of developing EDs [4]. In dancers, it is well known that they spend countless hours practicing in front of mirrors where their bodies are closely examined by themselves and others. In addition, high levels of perfectionism concerning dance and a specific body shape, combined with the socio-cultural pressures for thinness inherent in the dance profession and expectations of high performance, produce the ideal social climate for the development of EDs [5,6].

Similarly, the Tripartite Influence Model, which is an etiological model of body image disturbance and eating disorders, proposes that social agents, such as family, friends, and the media, promote ideals of appearance that emphasize a slim ideal for women and a lean, muscular ideal for men [7]. EDs today are a health problem in Western countries [8], and their incidence and prevalence are increasing. The prevalence of EDs in female athletes appears to be high when competitive weight is important [9]. This prevalence is higher in athletes (18%) compared to non-athletes (5%) [10], and is also higher in female athletes (20%) than in non-athletes (5%) [11]. The highest prevalence of EDs is found in aesthetic sports, as well as in sports where athletes are classified by body weight (45%) compared to other sports (12%) [11]. Similar to the trend in the prevalence of Eds, athletes in lean sports exhibit more disordered eating behaviors than those in non-lean sports [12]. Female dancers, in particular, show high levels of perfectionism and when in highly competitive environments, such as dance companies or professional conservatories, may present a higher risk for developing an ED [13,14]. EDs are characterized by chronicity and relapses of disordered eating behavior in which the attitudes of adolescent girls towards body weight, as well as their perception of body shape, are frequently altered. 

Numerous efforts have been made to develop instruments to improve the predictive value in the diagnostic screening of these diseases and various tests have been developed (EAT, Children’s Eating Attitudes Test (ChEAT), Eating Disorders Examination-Questionnaire (EDE-Q), Sick, Control, One, Fat, Food (SCOFF), Eating Disorder Inventory (EDI), etc.). Of all these questionnaires, the Eating Attitudes Test (EAT) has been the most extensively used because of its reliability and reproducibility in the detection of EDs in the general population [15]; it has also been widely used in sports and dance [16,17,18,19]. EDs are complex and have an impact on both the physical and social–emotional health of adolescents as well as young adults [20]. EDs such as anorexia nervosa and bulimia, have well-established effects on body composition, such as decreased fat mass, fat-free mass, and total body water [21], as well as decreased bone mineral density [22]. 

The somatotype is defined as the quantification of the present shape and composition of the human body. It is expressed as a three-number rating representing the endomorphy, mesomorphy, and ectomorphy components, respectively, always in the same order. Endomorphy is the relative fatness; mesomorphy is the relative musculoskeletal robustness; and ectomorphy is the relative linearity or slenderness of a physique. The Heath-Carter method uses various anthropometric measurements including weight, height, upper arm circumference, maximal calf circumference, femur and humerus breadths, and triceps, subscapular, supraspinal, and medial calf skinfolds [23]. Somatotype determination is useful in the characterization of body shape in contemporary dance and sports dance [24], but the relationship between anthropometric somatotype components and EDs has been rarely studied [25]. Mesomorphy, the second component of the anthropometric somatotype [23], reflects the development of skeleton and muscle tissues. Studies have found associations between body shape and muscularity and an increase in eating problems [26]. The rationale for studying the somatotype components and their association with DEA is that this measurement is a reflection of body shape based on the three components: endomorphy (fatness), mesomorphy (robustness), and ectomorphy (slenderness). In addition, body dissatisfaction seems to be a determining factor for risk behavior for EDs [2,26].

The present study therefore aimed to establish the accuracy of somatotype components using receiver operating characteristic (ROC) curve analysis to assess disordered eating attitudes (DEA) in a group of dance students engaged in beginner and advanced dance training. 

## 2. Methods

This cross-sectional and correlational study conducted in 2017 was approved by the Research Ethics Committee of the University of Málaga, Spain (EMEFYDE 2016–011 report) and carried out according to the principles of the Declaration of Helsinki. 

### 2.1. Participants

A total of 81 female students between the ages of 8 and 21 years participated in this study. All were enrolled in the Professional Conservatory of Granada, Spain, in courses from beginner training (BT) through advanced training (AT). The students in the AT group were distributed into four dance specialties: Flamenco, Spanish, Classical, and Contemporary. Participation in the study was voluntary, and prior to its initiation, written informed consent was obtained from the participants or the legal guardians of those under 18 years of age. The exclusion criteria were the inability to perform some of the anthropometric measurements, incorrectly completing the EAT-26, and male gender. 

### 2.2. Eating Behavior

Eating behavior was assessed with the EAT-26, which is a self-administered questionnaire used worldwide. It has been validated for assessing symptoms, concerns, and attitudes associated with abnormal eating behavior. The EAT-26 consists of 26 items forming three scales: dieting (related to the avoidance of fattening foods and the preoccupation with being thinner), bulimia and food preoccupation (involving items reflecting thoughts about food and those indicating bulimia), and oral control (associated with the self-control of eating and the perceived pressure from others to gain weight). A total score equal to or greater than 20 on the questionnaire is indicative of disordered eating behavior [15]. The EAT-26 has been validated for the Spanish population [16,27,28].

### 2.3. Anthropometric Assessment 

All anthropometric measurements were conducted after a 12-h fast. Weight was measured on a SECA 813 electronic scale (SECA, Hamburg, Germany) accurate to 0.1 kg., and stretch stature was measured using a wall-mounted SECA 216 stadiometer (SECA, Hamburg, Germany) accurate to 0.1 cm. Skinfolds were measured at the following sites: triceps, subscapular, supraspinal, and medial calf with a Holtain skinfold caliper (Holtain, Crymych, UK) accurate to 0.2 mm, computing the means for subsequent calculations. Girths were measured at the following sites: flexed and tensed arm and calf with a Lufkin W606PM anthropometric tape (Apex Tool Group, Lufkin, México) accurate to 0.1 cm. Biepicondylar humerus and Bicondylar femur breadths were measured with a Holtain sliding caliper (Holtain, Crymych, UK) accurate to 0.1 cm. BMI was calculated as weight in kilograms divided by height in meters squared. Anthropometric measurements were performed following standardized techniques adopted by the International Society for the Advancement of Kinanthropometry [29]. The technical error of measurement of the Level 3 anthropometrist was less than 3% for skinfolds and less than 1% for the rest of the anthropometric measurements.

### 2.4. Anthropometric Somatotype

Anthropometric somatotypes were determined according to the Heath-Carter method [23] by the following equations:Endomorphy = −0.7182 + 0.1451 (X) − 0.00068 (X^2^) + 0.0000014 (X^3^) where X = sum of triceps, subscapular and supraspinal skinfolds) multiplied by (170.18/height in cm).(1)

This is called height-corrected endomorphy and is the preferred method for calculating endomorphy.
Mesomorphy = 0.858 × humerus breadth + 0.601 × femur breadth + 0.188 × corrected arm girth + 0.161 × corrected calf girth − 0.131 × height + 4.5(2)

Three different equations are used to calculate ectomorphy according to the height–weight ratio (HWR). If the HWR is greater than or equal to 40.75, then ectomorphy = 0.732 HWR − 28.58. If HWR is less than 40.75 but greater than 38.25, then ectomorphy = 0.463 HWR − 17.63. If HWR is equal to or less than 38.25, then ectomorphy = 0.1.

### 2.5. Statistical Analysis

Normality was analyzed using the Shapiro-Wilk test. The descriptive characteristics of the group variables were expressed as mean ± standard deviation. Comparisons between groups were performed using a Mann-Whitney U or one-way ANOVA test when appropriate. Associations between EAT-26 subscale scores and somatotype components were assessed by Spearman’s rank correlation coefficient (rho) for the two groups separately. The following criteria were adopted to interpret the magnitude of the correlations: *r* ≤ 0.1 = trivial; 0.1 < *r* ≤ 0.3 = small; 0.3 < *r* ≤ 0.5 = moderate; 0. < *r* ≤ 0.7 = large; 0.7 < *r* ≤ 0.9 = very large; and *r* > 0.9 = almost perfect [30]. Cronbach’s alpha was performed for investigating the internal consistency of the subscales and the total score of the EAT-26 questionnaire. Test–retest reliability was assessed by intraclass correlation coefficient (ICC). The effect size was calculated using Rosenthal’s R-test and the power (1-β) to analyze the type II error by G*Power [31].

ROC curve analysis was used to test the performance of the somatotype components in predicting disordered eating status and to identify a cut-off score. The ROC curve is a graphical representation of a measure’s sensitivity plotted against its false positive rate (i.e., 1-specificity). The area under the curve (AUC) summarizes a test’s overall accuracy, or ability to distinguish cases from non-cases, based on the average value of sensitivity for all possible values of specificity. AUC values are defined as non-informative (≤0.50), less accurate (0.51 to 0.70), moderately accurate (0.71 to 0.90), highly accurate (0.91 to 0.99), or perfect (1.0) [32]. The likelihood ratios were calculated for each somatotype component. Binary logistic regression was used to evaluate the association between independent variables (somatotype components) and risk behaviors for DEA. The level of significance was set at *p* < 0.05. The statistical analysis was performed on MedCalc Statistical Software version 19.3.1 (MedCalc Software bvba, Ostend, Belgium).

## 3. Results

The risk of presenting DEA was 28.1% (*n* = 9) in the BT group and 6.1% (*n* = 3) in the AT group. Table 1 shows the comparative data of the study groups. Regarding the anthropometric variables; significant differences were found in age, weight, height, and BMI (all *p* = 0.000001). Similarly, differences were seen in both endomorphy (*p* = 0.003) and mesomorphy (*p* = 0.04). The psychometric assessment of the EAT-26 subscales showed differences between the groups only in bulimia (*p* = 0.004), although the values of the diet subscale and the total score were higher in the BT group (Table 1). 

The internal consistency of the EAT-26 was assessed with Cronbach’s alpha. The results revealed satisfactory levels for the subsample (*n* = 25). Cronbach’s alpha for the total EAT-26 score, bulimia, oral control, and dieting were 0.87, 0.915, 0.89, and 0.906, respectively. The subsample of 25 participants completed the retest after 3-week intervals. Test–retest reliability was good (ICC = 0.864, *p* < 0.001). A Mann–Whitney U test showed no significant differences between the test and re-test scores for the total EAT-26 score (*p* = 0.467).

Bivariate correlations in the beginner training group: bulimia correlated inversely with age (rho = −0.37, *p* = 0.035) and height (rho = −0.465, *p* = 0.007); oral control correlated with mesomorphy (rho = 0.46. *p* = 0.007); bulimia correlated inversely with height (rho = −0.39. *p*= 0.027) and directly with mesomorphy (rho = 0.40, *p* = 0.021); and the total score correlated inversely with height (rho = −0.45, *p* = 0.0089) and directly with mesomorphy (rho = 0.40, *p*= 0.02) (Table 2).

Bivariate correlations in the advanced training group: bulimia did not correlate with body composition variables or somatotype components (*p* > 0.05); oral control correlated directly with age (rho = 0.28, *p* = 0.047) and BMI (rho = 0.29, *p* = 0.04); the mesomorphic component correlated inversely with height (rho = −0.75, *p* < 0.0001) and directly with BMI (rho = 0.62, *p* < 0.0001); ectomorphy also correlated directly with diet (rho = 0.34, *p* = 0.016) and height (rho = 0.69, *p* < 0.001); and the total EAT-26 score did not correlate with any of the body composition variables or somatotype components (*p* > 0.05) (Table 3). 

### Analysis of ROC Curves 

Ectomorphy showed moderate accuracy for the diagnosis of DEA (AUC: 0.768, *p* = 0.0158), and mesomorphy showed moderately high accuracy (AUC: 0.82, *p* = 0.0003) in the BT group. Endomorphy in the BT group and all three components for the AT group indicated low accuracy (*p* > 0.05), (Figure 1 and Table 4).

The BT group had moderate specificity values for ectomorphy (78.6%) and high values for mesomorphy (95.6%) and endomorphy (87%). Sensitivity values were low and moderate, respectively. In this group, of note is the cut-off point of 6.34 for mesomorphy with a very high positive likelihood ratio value of (15.33). In the AT group, sensitivity values were moderate for endomorphy (67%) and highest for mesomorphy and ectomorphy (both 100%) (Table 5).

A logistic regression analysis was performed to examine independent associations between somatotype components and the probability of DEA (Table 6). The model provided a good fit for all three components in the BT group, particularly the mesomorphic component. However, in the AT group, none of the components achieved a significant fit.

## 4. Discussion

This study addresses the diagnostic accuracy of somatotype components (endomorphy, mesomorphy, and ectomorphy) to discriminate DEA in two groups of female dance students. The results show important significant differences and indicate that the ectomorphic and mesomorphic components in the BT group (younger students) can discriminate DEA with moderate and high accuracy, respectively; however, none of the somatotype components can do this in the AT group.

The emerging importance of somatotype components in predicting DEA in young female dance students is poorly reflected in the literature. Only one recent study showed that women who support attitudes and/or behaviors aimed at achieving the female muscle ideal may be susceptible to experiencing symptoms of DEA and negative emotional states, such as depression, stress, and anxiety [33]. 

The differences in the somatotype component associations between the study groups can be explained by the following reasons: the shorter height in the BT group increased mesomorphy scores, and although no significant differences in mesomorphy scores were found between groups, the levels were slightly higher in the BT group. In other words, the musculoskeletal structure was proportionally larger, and therefore a negative body image could have been internalized [34]. This also occurred with ectomorphy, since higher weight in relation to shorter height resulted in a lower ectomorphy value.

Similar results were found by Bartsch et al. who associate bulimia episodes with heavier somatotypes, such as pycnomorphic and metromorphic (similar to Heath and Carter’s endomorphic and mesomorphic components) [25]. In the present study, we also observed a relationship between the bulimia subscale and mesomorphy, but this was only in the BT group. It should be noted that all the subscales of the EAT-26 as well as the total score had a significant direct association with the mesomorphic component. The substantial increase in DEA in Western societies may be associated with sociocultural pressures to maintain a slim body shape as well as with the combined influence of perfectionism and learning [5]. This condition becomes more pronounced in aesthetic activities such as dance [35].

Investigating the nature and distribution of DEA in dance students is important for several reasons. First, these studies can provide information about the extent of DEA and weight control in female dancers. Second, on a broader level, many dancers face specific pressures to control their weight and body shape, and this enables us to explore the association between this type of pressure and the development of disordered eating attitudes and behaviors [36].

Although no reference was made to the prevalence of bulimia, anorexia nervosa, or non-specific disorders, due both to this not being an aim of the study and to the limited sample size, the prevalence of these disorders was higher in the BT group, comprising younger students, similar to results published by other authors. In the BT group, this prevalence was around 26% and in the AT group it was 6% [37].

In a study by Fortes et al. that sought to establish the extent to which anthropometric variables can explain body dissatisfaction and eating behavior disturbances in teenage boys and girls, the findings showed that body dissatisfaction was modulated by body fat percentage (R^2^ = 0.18), but this modulation was low. This was also low for its interaction with BMI (R^2^ = 0.18). The interaction between eating behavior disturbances and body fat percentage and between body fat percentage and BMI were both (R^2^ = 0.03) [38]. In contrast, Jáuregui et al. found that although there were correlations between BMI and EAT-40 scores, this did not result in an increased risk of developing an DEA [6]. These associations between BMI and the EAT-26 subscales were not confirmed in either of our study groups. The study by Toro et al. also found no significant inverse correlations between BMI and EAT-26 in dance students aged 14.4 years [39].

Another study in women found associations between BMI and DEA, and when these were mediated by a degree of body dissatisfaction, the associations disappeared [40]. This could explain the different associations between BMI and EAT-26 scores in our study, noting an association between BMI and oral control in the AT group, which implies greater self-control over food intake. In young people, being female, overweight, depressed, and having a high BMI are associated with DEA [41]. The results of a study of athletes competing in weight categories (taekwondo and judo) demonstrated that eating patterns and good dietary management decrease the likelihood of developing an DEA, which appears to be a good strategy [42].

Rouzitalab et al. analyzed a group of physical education students between the ages of 18 and 25 years. They found that EAT-26 had low but significant correlations with body weight and waist circumference in women, indicating these variables as good markers associated with an increase in DEA [43]. In our study, no correlations were found between weight or BMI and the total score in the AT group, and none of these correlations were found in the BT group. 

Several studies are available showing values for the subscales and total score on the EAT-26 in dance students of various ages, with no major differences found between these scores. These studies were performed in ballet dancers between the ages of 11 and 19, the same as those of our study. All the subscale scores in all the studies are higher in dieting than in oral control and bulimia [3,16,44,45]. 

Female gender, excess weight, living in urban areas, a distorted perception of weight, and body dissatisfaction were factors associated with DEA, suggesting that multiple factors contributed to the development of DEA [46,47]. The highest odds ratios were attributed to factors such as distorted perception of weight and body shape dissatisfaction [42,43]. In a study of 15- to 18-year-olds, the analysis showed significant negative associations between the total EAT-26 score and perceived physical appearance (rho = −0.290, *p* < 0.001) [37]. These data do not agree with those obtained in our study regarding correlations, although the EAT-26 score is very similar. However, for their assessment these differences should be compared with other determinants such as anxiety, stress, dissatisfaction, or body image and with other psychometric instruments that could explain those differences.

The results of this study can be used for prevention and early detection of DEA among young female dance students. Educational programs addressing adolescent eating behaviors should be developed. Future training programs on the significance of somatotype variations with growth, physical activity, and nutrition may be useful in preventing DEA.

## 5. Limitations

Several limitations of this study should be mentioned including the cross-sectional design, which limits the interpretation of causality. In addition, there was only one study stage, with no clinical interview performed to diagnose DEA. The EAT-26 was used as a screening tool only as the use of a questionnaire alone is not sufficient to diagnose DEA. Another limitation was that our study had a relatively small sample and only female students were included, although the statistical power was acceptable.

## 6. Conclusions 

The ROC curve analysis showed that the mesomorphy and ectomorphy components for the BT group had a high and moderate discriminatory power for classifying DEA. This study allows the identification of a population at risk of DEA, such as young female dance students. We believe that the results found in the group comprising the youngest dancers should serve as an alert for greater control in the onset of DEA.

## Figures and Tables

**Figure 1 jcm-09-02024-f001:**
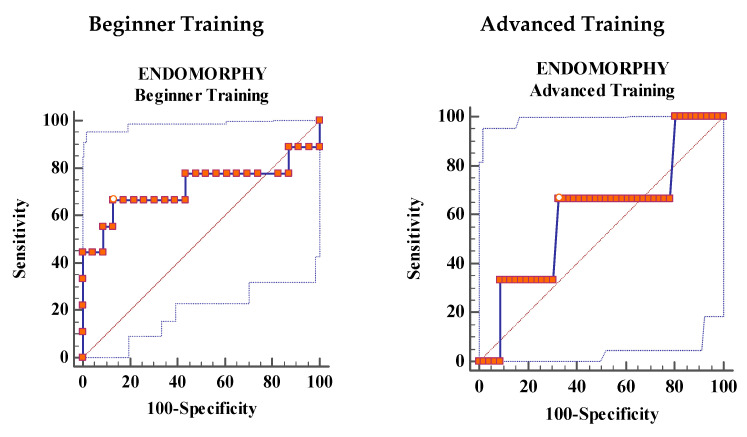
Receiver Operating Characteristics (ROC) curve analysis showing the area under the curve for the prediction of disordered eating attitudes using the Eating Attitudes Test-26 (EAT-26) total score.

**Table 1 jcm-09-02024-t001:** Descriptive data for anthropometrics and the Eating Attitudes Test-26 (EAT-26) subscales of the study groups.

Variables	Beginner Training		Advanced Training		*p*-Value	Effect Size	Statistical Power
	(*n* = 32)		(*n* = 49)				
	Mean	SD	Mean	SD		*r*	1-β
Age (years)	10.09	1.23	15.37	2.11	<0.0001	0.84	0.99
Weight (kg)	34.93	3.63	52.28	5.79	<0.0001	0.87	0.99
Height (m)	1.40	0.07	1.59	0.08	<0.0001	0.79	0.99
BMI (kg/m^2^)	18.00	1.89	20.69	1.96	<0.0001	0.57	0.99
Endomorphy	3.85	1.23	3.19	0.64	0.003	0.32	0.81
Mesomorphy	5.64	0.95	5.17	1.04	0.041	0.23	0.51
Ectomorphy	2.77	1.27	2.62	1.29	0.60	0.09	0.08
Bulimia	0.94	1.50	0.16	0.55	0.004	0.33	0.83
Oral control	3.44	4.06	3.10	3.05	0.66	0.05	0.07
Dieting	6.72	7.35	4.22	5.21	0.21	0.19	0.38
Total score	11.09	11.58	7.49	8.01	0.46	0.18	0.33

**Table 2 jcm-09-02024-t002:** Spearman’s rank correlation coefficient between body composition variables, somatotype components and the EAT-26 subscales in the Beginner Training group.

	Bulimia	Oral Control	Dieting	Total Score	Age	Weight	Height	BMI	Endo	Meso
**Oral control**	0.786 **									
**Dieting**	0.667 **	0.60 **								
**Total Score**	0.772 **	0.821 **	0.936 **							
**Age**	−0.37 *	−0.243	−0.158	−0.218						
**Weight**	−0.132	−0.212	−0.093	−0.17	0.54 **					
**Height**	−0.46 **	−0.48 **	−0.39 *	−0.45 **	0.81 **	0.528 **				
**BMI**	0.193	0.197	0.204	0.184	−0.169	0.537 **	−0.37 *			
**Endo**	0.254	0.206	0.116	0.121	−0.56 **	0.058	−0.65 **	0.63 **		
**Meso**	0.422 *	0.466 **	0.405 *	0.438 *	−0.166	0.32	−0.43 **	0.82 **	0.49 **	
**Ecto**	−0.319	−0.338	−0.27	−0.294	0.415 *	-0.291	0.62 **	−0.94 **	−0.73 **	−0.8 **

BMI: body mass index. Endo: endomorphy. Meso: mesomorphy. Ecto: ectomorphy. * *p* < 0.05. ** *p* < 0.001.

**Table 3 jcm-09-02024-t003:** Spearman’s rank correlation coefficients between body composition variables, somatotype components, and the EAT-26 subscales in the Advanced Training group.

	Bulimia	Oral Control	Dieting	Total Score	Age	Weight	Height	BMI	Endo	Meso
Oral control	0.469 **									
Dieting	0.489 **	0.141								
Total Score	0.482 *	0.828 **	0.616 **							
Age	0.23	0.284 *	−0.093	0.127						
Weight	0.237	0.35 *	−0.035	0.214	0.803 **					
Height	0.187	0.066	0.258	0.219	0.414 **	0.594 **				
BMI	0.103	0.294 *	−0.277	−0.011	0.474 **	0.51 **	−0.322 *			
Endo	−0.016	0.066	−0.107	−0.021	−0.165	−0.213	−0.57 **	0.431 **		
Meso	−0.09	0.102	−0.281	−0.112	−0.022	−0.146	−0.76 **	0.629 **	0.544 **	
Ecto	0.026	−0.177	0.342 *	0.119	−0.127	−0.118	0.692 **	−0.88 **	−0.59 **	−0.80 **

BMI, body mass index; Endo, endomorphy; Meso, mesomorphy; Ecto, ectomorphy. * *p* <0.05. ** *p* < 0.001.

**Table 4 jcm-09-02024-t004:** Characteristics of the ROC curves for somatotype components in the beginner training and advanced training groups.

	Beginner Training			Advanced Training		
	Endo	Meso	Ecto	Endo	Meso	Ecto
**Area under the curve**	0.72	0.82	0.768	0.601	0.522	0.558
**Standard error**	0.135	0.089	0.111	0.212	0.114	0.139
**95% CI**	0.53 to 0.86	0.64 to 0.93	0.58 to 0.89	0.45 to 0.74	0.37 to 0.66	0.41 to 0.70
**z statistic**	1.631	3.61	2.41	0.477	0.191	0.418
***p*** **-value**	0.1030	0.003	0.0158	0.633	0.8483	0.6758
**Youden’s J index**	0.5362	0.6232	0.564	0.3406	0.3696	0.3696

ROC, receiver operating characteristic; Endo, endomorphy; Meso, mesomorphy; Ecto, ectomorphy; CI, confidence interval.

**Table 5 jcm-09-02024-t005:** Sensitivity. specificity and likelihood ratios of somatotype components in the beginner and advanced training groups.

Somatotype Component	Training	Cut-Off	Sens	95% CI	Spec	95% CI	+LR	95% CI	−LR	95% CI
**Endomorphy**	BT	>4.36	66.67	29.9–92.5	86.96	66.4–97.2	5.11	1.6–16.2	0.38	0.2–1.0
	AT	≤2.88	66.67	9.4–99.2	67.39	52.0–80.5	2.04	0.8–5.0	0.49	0.10–2.5
**Mesomorphy**	BT	>6.34	66.67	29.9–92.5	95.65	78.1–99.9	15.33	2.1–110	0.35	0.1–0.9
	AT	≤5.72	100	29.2–100.0	36.96	23.2–52.5	1.59	1.3–2.0	0	
**Ectomorphy**	BT	≤2.41	77.78	40.0–97.2	78.6	56.3–92.5	3.58	1.5–8.4	0.28	0.08–1.0
	AT	>1.76	100	29.2–100.0	36.96	23.2–52.5	1.59	1.3–2.0	0	

BT, beginner training; AT, advanced training; Sens, sensitivity; CI, confidence interval; Spec, specificity; +LR, positive likelihood ratio; −LR, negative likelihood ratio.

**Table 6 jcm-09-02024-t006:** Associations between somatotype components and eating disorders.

				Overall Model Fit		Hosmer & Lemeshow	
Group	Variable	OR	95% CI	*X* ^2^	*p*-Value	*X* ^2^	*p*-Value
BT	Endo	2.44	1.1608 to 5.1493	6.878	0.0087	7.3902	0.5966
	Meso	7.14	1.7214 to 29.672	12.47	0.0004	11.0945	0.2693
	Ecto	0.42	0.2041 to 0.8972	6.22	0.0126	9.5698	0.3864
AT	Endo	Not retained in the model					
	Meso	Not retained in the model					
	Ecto	Not retained in the model					

BT, beginner training; AT, advanced training; Endo, endomorphy; Meso, mesomorphy; Ecto, ectomorphy; OR, odds ratio; CI, confidence interval; *X*^2^, chi-squared.

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
