# Peer review of "Somatotype Components as Useful Predictors of Disordered Eating Attitudes in Young Female Ballet Dance Students"

_jcm, 2020, doi:10.3390/jcm9072024_

Round 1

Reviewer 1 Report

the authors of the present paper aimed at  examining  the accuracy of somatotype components in correctly classifying eating disorders in female dance students by using the receiver operating characteristic curve analysis.

The paper is of general interest. However, there are few points inside the manuscript that should be clarified and it should be corrected by a native speaking English. The organization and presentation of the paper is very very bad. The manuscript is poorly written, very confuse and sloppy. For example authors mentioned, in the abstract, Basic Teachings (BT) and Professional Teachings (PT); in the methods they became beginner training (BT) through advanced training (AT). Line 39 Toro et al., 2006, should be a number for the reference.

But, my main concern regards the two analyzed groups in terms of age. Mean age of BT group was 10 and mean age of PT group is 15; the BT group is in the childhood stage, PT group is in the stage of pre-adolescence and adolescence. This means that the problems are very different from one group to another, for example in the hormonal response.

A better description of the dance students should be added, including the diet followed by the athletes.

Moreover, it should be important to add to the paper whether the sample size is adequate (power calculation), authors say that a limitation of the paper is the small sample size.

In addition, at least to me, the manuscript is very hard to follow in some part.

Author Response

Comments and Suggestions for Authors

Answers

the authors of the present paper aimed at  examining  the accuracy of somatotype components in correctly classifying eating disorders in female dance students by using the receiver operating characteristic curve analysis. The paper is of general interest.

Thank you for your suggestions, which have undoubtedly served to improve our manuscript.

However, there are few points inside the manuscript that should be clarified and it should be corrected by a native speaking English.

The manuscript has been revised by a native English-speaking translator.

The manuscript is poorly written, very confuse and sloppy. For example authors mentioned, in the abstract, Basic Teachings (BT) and Professional Teachings (PT); in the methods they became beginner training (BT) through advanced training (AT).

The denominations of the groups in the abstract have been corrected (beginner and advanced training groups).

line 39 Toro et al., 2006, should be a number for the reference.

This mistake has been corrected.

But, my main concern regards the two analyzed groups in terms of age. Mean age of BT group was 10 and mean age of PT group is 15; the BT group is in the childhood stage, PT group is in the stage of pre-adolescence and adolescence. This means that the problems are very different from one group to another, for example in the hormonal response.

The study examined ballet students at a professional dance conservatory in Spain at different stages of training. There are age differences because the beginners (BT), or the younger age group, begin lessons as early as 8 years of age. The advanced level (AT), which lasts 6 years, is begun only after completion of 4 years of BT.

Clearly, there will be hormonal as well as other differences, but the objective of this study was to examine the relationships between the somatotype components and EDs and the accuracy of these components to predict EDs.

A better description of the dance students should be added, including the diet followed by the athletes.

Unfortunately, this information was not collected.

Moreover, it should be important to add to the paper whether the sample size is adequate (power calculation), authors say that a limitation of the paper is the small sample size.

The power calculation was performed by G*Power. This information has been included in Table 1.

In addition, at least to me, the manuscript is very hard to follow in some part.

The order of certain parts of the text has been modified to make it clearer and more easily understood as well as to include the comments and recommendations of the other reviewers.

Reviewer 2 Report

Thank you for the opportunity to review the manuscript entitled, “Somatotype components as useful predictors of eating disorders in young female ballet dance students”This manuscript details a cross-sectional study investigating the relationship between somatotype components and disordered eating symptoms in 81 children and adolescent female ballet dancers. The paper seeks to advance the field's knowledge on eating disorder cognitions and behaviors by analyzing self-report questionnaires and anthropometrics in youth across two levels of ballet experience. 

Several areas of concern are noted throughout the manuscript in its present form. As written, the introduction is too cursory an overview of this topic and leaves out important information related to a variety of eating disorder diagnoses. For instance, the authors mention bulimia nervosa on line 36 but do not provide any further context. The authors have not included Avoidant Restrictive Food Intake Disorder (ARFID) in their overview, which is recognized as a disorder in the DSM-5. It is also recommended that the sentence referencing the etiology of eating disorders be removed or significantly revised, as the scope is far too narrowly presented. It is misleading to imply that personality factors alone contribute to the risk of developing an eating disorder. Further, eating disorders are present throughout the world, across various cultures in developed and non-developed countries as well as Western and non-Western cultures. The paper would also be strengthened if additional background was provided when introducing the term "somatotype components" in the introduction to provide enough context for readers who might be unfamiliar with the construct. The background section also does not appear to sufficiently justify the study design and should be expanded.

Additional concerns include the choice of the EAT-26 instead of the ch-EAT for the young participants in the sample; rationale and citations supporting the validation of the EAT-26 in young children would strengthen this choice. It is also recommended to include internal consistency statistics for the measures used in this study. 

The use and interpretation of the ROC analyses is confusing as written. Additional information regarding how the logistic regression analyses were conducted is also needed. From Table 1, it appears that no participants in the study scored greater than a 20 on the EAT-26; as such, it is unclear how the probability of an ED was defined in the logistic regression analyses. Table 7 is also not needed and can be removed. Median values are presented in Table 1 whereas mean values are presented in Table 7; please be consistent throughout manuscript. It is also unclear why height and EAT-26 symptoms were correlated - is this not confused by age or puberty status? 

In the discussion, the authors discuss the prevalence of eating disorder diagnoses in their sample; how was this information collected? Which eating disorder diagnoses were included? Please consider adding additional information to clarify. 

One minor point - please consider updating Tables 2 and 3 with asterisks next to the significant findings instead of listing the p-values for each correlation. This will improve readability and highlight significant findings.

Author Response

Comments and Suggestions for Authors

Answers

Thank you for the opportunity to review the manuscript entitled, “Somatotype components as useful predictors of eating disorders in young female ballet dance students”This manuscript details a cross-sectional study investigating the relationship between somatotype components and disordered eating symptoms in 81 children and adolescent female ballet dancers. The paper seeks to advance the field's knowledge on eating disorder cognitions and behaviors by analyzing self-report questionnaires and anthropometrics in youth across two levels of ballet experience

Thank you for your suggestions, which have undoubtedly served to improve our manuscript. The main objective was to study the predictive capacity of the somatotype components to discriminate the presence of EDs.

The manuscript has been revised by a native English-speaking translator.

Several areas of concern are noted throughout the manuscript in its present form. As written, the introduction is too cursory an overview of this topic and leaves out important information related to a variety of eating disorder diagnoses. For instance, the authors mention bulimia nervosa on line 36 but do not provide any further context. The authors have not included Avoidant Restrictive Food Intake Disorder (ARFID) in their overview, which is recognized as a disorder in the DSM-5.

A broader and more accurate explanation has now been included in the Introduction based on a classification of the EDs.

It is also recommended that the sentence referencing the etiology of eating disorders be removed or significantly revised, as the scope is far too narrowly presented.

This sentence has been changed and improved.

It is misleading to imply that personality factors alone contribute to the risk of developing an eating disorder. Further, eating disorders are present throughout the world, across various cultures in developed and non-developed countries as well as Western and non-Western cultures.

Other factors have been included in the Introduction. More information has been given (e.g. epigenetic causes are also presented).

The paper would also be strengthened if additional background was provided when introducing the term "somatotype components" in the introduction to provide enough context for readers who might be unfamiliar with the construct

To provide further information for the reader, a definition of the concept of somatotype has been included in the Introduction.

Additional concerns include the choice of the EAT-26 instead of the ch-EAT for the young participants in the sample; rationale and citations supporting the validation of the EAT-26 in young children would strengthen this choice.

Smolak and Levine studied the EAT26 vs the ChEAT: As is true of the EAT, the ChEAT is considered more as a screening and research instrument rather than a diagnostic tool. Indeed, there is no evidence that the ChEAT can identify diagnostic groups. Research with clinical samples is required to address that issue. Given the low rate of childhood eating disorders, however, the determination of how well ChEAT scores predict or correlate with adolescent diagnostic categories and treatment outcomes is a more pressing next step in this line of research.

It is also recommended to include internal consistency statistics for the measures used in this study. 

This information can be seen in the Methods (Chronbach’s alpha and ICC).

The use and interpretation of the ROC analyses is confusing as written.

Some characteristics of ROC curves are expressed in the text, mainly Sensitivity and Specificity. For more information, Tables 4 and 5 and Figure 1 are provided.

Additional information regarding how the logistic regression analyses were conducted is also needed. From Table 1, it appears that no participants in the study scored greater than a 20 on the EAT-26; as such, it is unclear how the probability of an ED was defined in the logistic regression analyses.

A value of >20 is the discriminatory value for EDs in the Spanish population.

 (Peláez-Fernández, M.M.; Ruiz-Lázaro, P.M.; Labrador, F.J.; Raich, R.M. Validation of the Eating Attitudes Test as a screening instrument for eating disorders in general population. Med. Clin. (Barc). 2014. 20. 153–155)

Table 7 is also not needed and can be removed. Median values are presented in Table 1 whereas mean values are presented in Table 7; please be consistent throughout manuscript.

Table 7 has been removed. This information has been introduced in the Discussion.

It is also unclear why height and EAT-26 symptoms were correlated - is this not confused by age or puberty status? 

Your comment is opportune. The finding is considered important as it links to the concept of somatototype, where inverse associations of all the subscales were found with the height of the younger age group of girls, who likely have a distortion of their body figure and body shape.   

In the discussion, the authors discuss the prevalence of eating disorder diagnoses in their sample; how was this information collected? Which eating disorder diagnoses were included? Please consider adding additional information to clarify.

The Discussion is based on the references, as is logical, of Dotti, Toro, Arcelus and Hayakawa, in which values of some of the subscales and in all cases are compared with the total score.

One minor point - please consider updating Tables 2 and 3 with asterisks next to the significant findings instead of listing the p-values for each correlation. This will improve readability and highlight significant findings.

The requested changes have been made for improved readability of Tables 2 and 3.

Reviewer 3 Report

Authors investigated the feasibility of somatotype components for predicting (or detecting?) the risk of eating disorders. They found mesomorphy to have moderate-high accuracy, and ectomorphy to have moderate accuracy for detecting ED, only in a group of younger ballet dancers.  No much value was found on somatotype components for detecting ED on professional ballet dancers. The topic is relevant and interesting, but the manuscript requires modifications to improve clarity.

  • I suggest synthetizing more the background on EDs and its relevance (Introduction section). For example, if the approach used in this study was the somatotype components, more justification on such approach should be included, and be more concise on covering other approaches (social, psychological, etc.). In fact, sompatotype components are not defined, and the components are numbered starting with the “second component” (line 73 page 3).
  • You claim that “the relationship between anthropometric somatotype components and EDs has not been studied”, but then make confusing statements about “works that found relationships of body shape and muscularity with more eating problems”, and “heavier somatotypes were associated with a higher frequency of bulimic disorders”. Are not eating problems and bulimic disorders considered EDs?  A better definition of these terms, relations, and relationships is required.
  • Some justification and rationale for using ROC curves analysis is also required.
  • A detailed definition (and justification) of somatotype components, as well as the other measurements taken of the study, is required. It was not clear to me, for example, why did they measure skinfolds, girths, etc. Are they part of the somatotype components?
  • Later in the methods the mathematical definitions appear (the components are finally listed in the discussion as endomorphy, mesomorphy, and ectomorphy), but the structure of the paper is confusing in the overall. I suggest clearly defining the terms before being used.
  • Did you try combining the groups to assess overall accuracy? The results are not good in professional ballet dancers, and further analysis like machine learning classification could be explored to better evaluate the feasibility of somatotype components for prediction of EDs. 
  • The paper is well written in the overall, but some mistakes must be revised. One example: “EDs today are a health problem specific to developed countries, and their incidence and prevalence are increasing with a prevalence of EDs in female athletes appears to be high when competitive weight is important”

Author Response

Comments and Suggestions for Authors

Answers

Authors investigated the feasibility of somatotype components for predicting (or detecting?) the risk of eating disorders. They found mesomorphy to have moderate-high accuracy, and ectomorphy to have moderate accuracy for detecting ED, only in a group of younger ballet dancers.  No much value was found on somatotype components for detecting ED on professional ballet dancers. The topic is relevant and interesting, but the manuscript requires modifications to improve clarity.

Thank you for your suggestions, which have undoubtedly served to improve our manuscript.

The manuscript has been revised by a native English-speaking translator.

I suggest synthetizing more the background on EDs and its relevance (Introduction section). For example, if the approach used in this study was the somatotype components, more justification on such approach should be included, and be more concise on covering other approaches (social, psychological, etc.).

This section has been modified.

In fact, sompatotype components are not defined, and the components are numbered starting with the “second component” (line 73 page 3).

Somatotype has been defined in the Introduction.

You claim that “the relationship between anthropometric somatotype components and EDs has not been studied”, but then make confusing statements about “works that found relationships of body shape and muscularity with more eating problems”, and “heavier somatotypes were associated with a higher frequency of bulimic disorders”. Are not eating problems and bulimic disorders considered EDs?  A better definition of these terms, relations, and relationships is required

Only one study was found in which muscularity is associated (by other methods such as Strömgren's Metrik Index) with EDs.

(Bartsch AJ, Brümmerhoff A, Greil H, Neumärker KJ. Shall the anthropometry of physique cast new light on the diagnoses and treatment of eating disorders? Eur Child Adolesc Psychiatry. 2003;12(S1):i54-i.)

Some justification and rationale for using ROC curves analysis is also required.

ROC plots provide a statistical method to assess the diagnostic accuracy of a test (or biomarker) that has a continuous spectrum of test results. The ROC curve is a graphical display of the trade-offs of the true-positive rate (sensitivity) and the false-positive rate (1-specificity) corresponding to all possible binary tests that can be formed from this continuous variable. Each classification rule, or cut-off level, generates a point on the graph. The closer the curve follows the left-hand border and then the top border of the ROC space, the more accurate the test.

A detailed definition (and justification) of somatotype components, as well as the other measurements taken of the study, is required. It was not clear to me, for example, why did they measure skinfolds, girths, etc. Are they part of the somatotype components?

The anthropometric somatotype is obtained by means of ten measurements, including skinfolds, girths and breadths (see Introduction).

Later in the methods the mathematical definitions appear (the components are finally listed in the discussion as endomorphy, mesomorphy, and ectomorphy), but the structure of the paper is confusing in the overall. I suggest clearly defining the terms before being used.

The concepts have been defined in the Introduction to provide greater knowledge and understanding for the reader.

Did you try combining the groups to assess overall accuracy? The results are not good in professional ballet dancers, and further analysis like machine learning classification could be explored to better evaluate the feasibility of somatotype components for prediction of EDs

Yes, this was done as well, but the differences found made the differentiated analysis by groups (Beginner and Advanced)

.The paper is well written in the overall, but some mistakes must be revised. One example: “EDs today are a health problem specific to developed countries, and their incidence and prevalence are increasing with a prevalence of EDs in female athletes appears to be high when competitive weight is important”

This sentence has been modified.

Round 2

Reviewer 1 Report

I congratulate the authors for the improvements made to the article.

I still have some concerns about the different age of the enrolled subjects, otherwise the article is suitable for publication.

Author Response

Rw1

Open Review

English language and style

( ) Extensive editing of English language and style required
( ) Moderate English changes required
(x) English language and style are fine/minor spell check required
( ) I don't feel qualified to judge about the English language and style

Comments and Suggestions for Authors

Comments and Suggestions for Authors

Answer

I congratulate the authors for the improvements made to the article.

I want to thank the expert reviewer for his constructive comments.

I still have some concerns about the different age of the enrolled subjects, otherwise the article is suitable for publication.

The study examined ballet students at a professional dance conservatory in Spain at different stages of training. There are age differences because the beginners (BT), or the younger age group, begin lessons as early as 8 years of age. The advanced level (AT), which lasts 6 years, is begun only after completion of 4 years of BT.

Reviewer 2 Report

The authors are commended for their efforts to revise the manuscript and address the reviewer's comments. The changes have substantially improved readability and strengthened the paper. The detailed comments in response to reviewer suggestions is also appreciated. 

A few concerns persist regarding the manuscript in its current form. 

  1. In the abstract, please clarify the first sentence in the results section. From the wording, it appears that the authors assessed and measured clinical eating disorder diagnoses, rather than disordered eating behaviors. The paper does not describe how the rates of eating disorder diagnoses were assessed in the study; if the results are indeed referring to the scores on the EAT-26, the abstract should be reflected to indicate that. Disordered eating attitudes or behaviors is likely more appropriate than the term eating disorder. Far too often, readers only read the abstract and may misconstrue the findings. Please update this distinction throughout the manuscript for clarity. Also please be mindful to define abbreviations when used for the first time. 
  2. In the introduction, the authors state that amenorrhea is a diagnostic criteria for Anorexia Nervosa. This is an outdated criteria from DSM-IV-TR which was removed in the DSM-5. Please revise accordingly. 
  3. The diagnosis "EDNOS" is also an outdated term. It was replaced with "Other Specified Feeding and Eating Disorder" in 2013. 
  4. It is not clear how the percentages in line 172 were determined. Was this from the number of participants who scored higher than 20 on the EAT-26? If so, this should be stated. 
  5. The discussion section states that somatotype components discriminated eating disorders in the female dance students. Should this be disordered eating behaviors or abnormal eating behavior, rather than eating disorders (diagnosis)? The EAT-26 is not enough to clinically diagnose eating disorders alone. 

Thank you for the opportunity to review this revised manuscript. 

Author Response

Rw 2

Open Review

English language and style

( ) Extensive editing of English language and style required
( ) Moderate English changes required
(x) English language and style are fine/minor spell check required
( ) I don't feel qualified to judge about the English language and style

Yes

Can be improved

Must be improved

Not applicable

Does the introduction provide sufficient background and include all relevant references?

( )

(x)

( )

( )

Is the research design appropriate?

(x)

( )

( )

( )

Are the methods adequately described?

(x)

( )

( )

( )

Are the results clearly presented?

( )

(x)

( )

( )

Are the conclusions supported by the results?

( )

( )

(x)

( )

Comments and Suggestions for Authors

Answer

The authors are commended for their efforts to revise the manuscript and address the reviewer's comments. The changes have substantially improved readability and strengthened the paper. The detailed comments in response to reviewer suggestions is also appreciated

I want to thank the expert reviewer for his constructive comments.

A few concerns persist regarding the manuscript in its current form. 

In the abstract, please clarify the first sentence in the results section. From the wording, it appears that the authors assessed and measured clinical eating disorder diagnoses, rather than disordered eating behaviors. The paper does not describe how the rates of eating disorder diagnoses were assessed in the study; if the results are indeed referring to the scores on the EAT-26, the abstract should be reflected to indicate that. Disordered eating attitudes or behaviors is likely more appropriate than the term eating disorder. Far too often, readers only read the abstract and may misconstrue the findings. Please update this distinction throughout the manuscript for clarity. Also please be mindful to define abbreviations when used for the first time

Indeed, the prevalence of DEA (disordered eating attitudes) was obtained from the EAT-26 test scores ≥ 20.

The term eating disorders has been changed to disordered eating attitudes in the abstract. We agree this is a more appropriate term and believe the abstract is now clear.

We have ensured that all abbreviations are defined at first use.

In the introduction, the authors state that amenorrhea is a diagnostic criteria for Anorexia Nervosa. This is an outdated criteria from DSM-IV-TR which was removed in the DSM-5. Please revise accordingly. 

The word “amenorrhea” has been removed from the text on P2 L 38.

The diagnosis "EDNOS" is also an outdated term. It was replaced with "Other Specified Feeding and Eating Disorder" in 2013

EDNOS has been removed and has been replaced with “Other Specified Feeding and Eating Disorders” on P2 L41.

It is not clear how the percentages in line 172 were determined. Was this from the number of participants who scored higher than 20 on the EAT-26? If so, this should be stated

Indeed, the prevalence of DEA (disordered eating attitudes) was obtained from the EAT-26 test scores ≥ 20.

This is stated in the abstract and in methods on P5 L118.

The discussion section states that somatotype components discriminated eating disorders in the female dance students. Should this be disordered eating behaviors or abnormal eating behavior, rather than eating disorders (diagnosis)? The EAT-26 is not enough to clinically diagnose eating disorders alone. 

The term eating disorders has been replaced with disordered eating attitudes where appropriate throughout the text.

Thank you for the opportunity to review this revised manuscript. 

I want to thank the expert reviewer for his constructive comments.

Reviewer 3 Report

The authors have addressed all my comments and suggestions.  

Author Response

Rw3

Open Review

English language and style

( ) Extensive editing of English language and style required
( ) Moderate English changes required
(x) English language and style are fine/minor spell check required
( ) I don't feel qualified to judge about the English language and style

Yes

Can be improved

Must be improved

Not applicable

Does the introduction provide sufficient background and include all relevant references?

(x)

( )

( )

( )

Is the research design appropriate?

(x)

( )

( )

( )

Are the methods adequately described?

(x)

( )

( )

( )

Are the results clearly presented?

( )

(x)

( )

( )

Are the conclusions supported by the results?

(x)

( )

( )

( )

Comments and Suggestions for Authors

Comments and Suggestions for Authors

Answer

The authors have addressed all my comments and suggestions.  

I want to thank the expert reviewer for his constructive comments.
